# Pharmacogenomics of Old and New Immunosuppressive Drugs for Precision Medicine in Kidney Transplantation

**DOI:** 10.3390/jcm12134454

**Published:** 2023-07-03

**Authors:** Stefano Turolo, Alberto Edefonti, Marie Luise Syren, Giovanni Montini

**Affiliations:** 1Fondazione IRCCS Ca’ Granda Ospedale Maggiore Policlinico, Pediatric Nephrology, Dialysis and Transplant Unit, 20122 Milan, Italy; alberto.edefonti@policlinico.mi.it (A.E.); giovanni.montini@unimi.it (G.M.); 2Department of Clinical Sciences and Community Health, University of Milan, 20122 Milan, Italy; eva.syren@unimi.it

**Keywords:** pharmacogenomics, cyclosporine, tacrolimus, CYP, mycophenolate mofetil

## Abstract

Kidney transplantation is the preferred therapeutic option for end-stage kidney disease, but, despite major therapeutic advancements, allograft rejection continues to endanger graft survival. Every patient is unique due to his or her clinical history, drug metabolism, genetic background, and epigenetics. For this reason, examples of “personalized medicine” and “precision medicine” have steadily increased in recent decades. The final target of precision medicine is to maximize drug efficacy and minimize toxicity for each individual patient. Immunosuppressive drugs, in the setting of kidney transplantation, require a precise dosage to avoid either adverse events (overdosage) or a lack of efficacy (underdosage). In this review, we will explore the knowledge regarding the pharmacogenomics of the main immunosuppressive medications currently utilized in kidney transplantation. We will focus on clinically relevant pharmacogenomic data, that is, the polymorphisms of the genes that metabolize immunosuppressive drugs.

## 1. Introduction

### 1.1. Allograft Rejection and Immunosuppressive Drugs

Allogenic kidney transplantation is the preferred therapeutic option for end-stage kidney disease. Despite progress in transplant medicine, including the availability of new immunosuppressive (IS) drugs and improved therapeutic protocols, acute and chronic allograft rejection continue to endanger allograft survival.

Allograft rejection is a sterile inflammation, caused by the innate immune response, induced independently of the adaptive immune response [1,2]. Four main mechanisms of rejection (hyperacute, acute, chronic, and combined acute and chronic) are presently recognized: (a) hyperacute rejection, caused by anti-donor antibodies that already existed in the recipient before transplantation, (b) acute rejection, driven by two main mechanisms, the cellular-mediated and the humoral (or antibody)-mediated, (c) chronic rejection, mediated by either humoral or cellular mechanisms, linked to memory/plasma cells and antibodies and representing the main cause of graft loss [3,4], (d) finally, a combination of acute and chronic rejection, has also been described [5].

There are several kinds of immunosuppressive drugs (IS), including calcineurin inhibitors, glucocorticoids, mTOR inhibitors, monoclonal and polyclonal antibodies, and antimetabolites, whose mechanisms and sites of action are summarized in Figure 1.

Like all drugs, IS agents are metabolized by a set of enzymes, such as cytochrome P-450, encoded by the gene cluster of CYP genes, or extruded from the cells by membrane transport proteins, such as the p-glycoprotein, encoded by the ABCB1 gene.

The mechanisms of metabolization of IS drugs affect their ability to prevent rejection. In turn, the activity of metabolizing enzymes depends on both gene expression and genetic polymorphisms.

Basiliximab acts as an IL-2 receptor agonist; sirolimus and everolimus inhibit cell replication by binding mTOR; tacrolimus and cyclosporine inhibit calcineurin by binding FKBP and cyclophilin, respectively; corticosteroids inhibit NFKB activity and azathioprine and mycophenolate inhibit cell cycle.

### 1.2. Pharmacogenomics for Precision Medicine

The concept of “personalized medicine” and “precision medicine” is that every patient is unique and should benefit from personalized therapy, because of their own metabolism, clinical history, genetic background, and epigenetic factors.

The final target of precision medicine for IS in transplantation is to maximize drug efficacy and minimize toxicity for each individual patient, avoiding overdosage and its consequent major adverse events such as infections, nephrotoxicity, or organ toxicity, and underdosage and the consequent occurrence of graft rejection.

Pharmacogenetics was the initial field of study for individual variation in drug response and was focused on the study of monogenic traits such as single nucleotide polymorphisms (SNPs) involved in drug metabolism.

Further pharmacogenetic studies increasingly involved the entire “pathways” of drug-gene interaction, including both factors that influence the concentration of a drug (pharmacokinetics) and factors associated with the drug’s bio-availability (pharmacodynamics).

Following the rapid technological developments in human genomics [6] pharmacogenetics evolved into pharmacogenomics.

With the new and continuous progress in understanding the molecular basis of drug action and the genetic determinants of drug response, pharmacogenomics aims to identify individuals who will respond to a particular drug treatment compared to those who have a low probability of response [7].

Unfortunately, not much of all vested knowledge has been translated into clinical practice and at the present time most drug-gene associations that showed some evidence of clinical validity have not progressed to clinical settings [8].

Despite these limits, pharmacogenomics is a very promising field of clinical research and in the future, it will be able to allow, through sophisticated algorithms, an individualized IS dosage. However, the future translation of pharmacogenomics into “personalized medicine” will depend on many factors, including clinical relevance, environmental-genetic interactions, and cost [9].

In this review, we will explore the present knowledge regarding the pharmacogenomics of the main IS medications currently utilized in kidney transplantation.

## 2. Methods

For this review, PubMed (www.pubmed.gov) was consulted in August 2022. No limit was given regarding the date of publication of the articles, and the following keywords were used: kidney transplantation, immunosuppressive drugs, cyclosporine, tacrolimus, everolimus, sirolimus, mycophenolate mofetil, rituximab, glucocorticoids, thiopurines, basiliximab and daclizumab.

For a more precise selection, we used the keywords above in combination with the following secondary set of keywords: genetic polymorphism, pharmacogenetic, and pharmacogenomics. In this way we obtained a prompt such as (kideney trtansplantation) AND (drug name) AND (key word of the second set).

Following the methodology described by Urzi Brancati et al. [10], we considered all articles, written in English, where an analysis was made of the role of genetics in IS drug bioavailability and metabolization. We excluded redundant articles and when results showed also another kind of transplantation we privileged articles dealing with kidney transplantation. Discussion between the authors regarding the selection of articles to be included in the review was then made, based on the quality of the publications.

The checklist proposed by the PRISMA statement was not applicable to this review, due to the nature of the topic, which is mainly bio-molecular instead of clinical. Consequently, several critical points of the PRISMA checklist could not be addressed. We considered the PICO approach where the population and problem are transplanted patients and immunosuppressive drug metabolization, the intervention is the treatment with different kinds of immunosuppressive therapy, Comparison is the role of the different polymorphisms and the Outcome is how they metabolize each drug.

## 3. Results

### 3.1. Calcineurin Inhibitors

#### 3.1.1. Cyclosporine

Cyclosporine A (CsA) is one of the oldest and still used immunosuppressive drugs. While it is no longer used for transplantation in high-income countries, it is still used in low-and medium-income countries.

Cyclosporine has as common side effects dose related to high blood pressure, numbness or tingling of the hands or feet, and swollen or inflamed gums.

The mechanism of action of the drug has received much attention. It has been clarified that, once arrived in the T-cell cytoplasm, CsA binds cyclophilin, which starts a series of events. The CsA-cyclophilin complex inhibits the phosphatase activity of calcineurin, which regulates nuclear translocation and subsequent activation of the Nuclear Factor of Activated T-cells (NFAT) transcription factors. In addition to the calcineurin/NFAT pathway, it has been recently reported that CsA can also block the activation of c-Jun N-terminal kinase (JNK) and p38 signaling pathways, triggered by antigen recognition [11].

Despite several enzymes, receptors, and pathways being involved in CsA mechanisms of action, none of their associated genes showed to be relevant in CsA pharmacogenomics, while factors involved in drug metabolization, such as membrane P-glycoprotein (encoded by ABCB1 gene) and cytochrome P450 (encoded by CYP cluster gene) were reported to play an important role. This is true, in particular, for the P450 enzyme encoded by the allotype CYP3A (in turn encoded by the gene CYP3A), which is present mainly in the human liver and small intestine and catalyzes the metabolism of structurally diverse xenobiotics [12,13].

The main genes involved in CsA pharmacogenomics are illustrated in the following paragraphs.

#### 3.1.2. CYP3A5 Gene

There is not a unanimous consensus in the literature as to the role of this gene in CsA metabolism. The study by Dan-ying [14] indicated CYP3A5 polymorphism as the main cause of the inter-individual CsA blood level variability. This data was also confirmed by another study [15], which showed that transplanted patients with the active form of the CYP3A5 gene required a higher CsA dosage. However, another study found no significant correlation between CsA through concentration or dose requirement and CYP3A5*1 genotype [16].

#### 3.1.3. CYP3A4 Gene

A full consensus has not been reached so far regarding the role of CYP3A4. It has been shown that transplanted patients with CYP3A4*1B polymorphism require a higher CsA dose compared to patients with the CYP3A4*1 allele [15]. Indeed, the former resulted in a higher average drug concentration in the blood [17,18], but other studies did not confirm this relationship [19], possibly due to the presence of unknown polymorphisms, or to the role of other genes, such as SXR [20].

Other CYP3A4 polymorphisms play a role in the pharmacodynamics of CsA: a Jordanian study found a strong direct correlation between the CYP3A4*22 polymorphism and a higher C0/dose [21].

#### 3.1.4. Membrane P-glycoprotein—ABCB1 Gene

As regards ABCB1, which is the most studied gene involved in the CsA metabolization, polymorphisms in exons 12 (C1236T), exon 21 (G2677T/A) and exon 26 (C3435T) were shown to influence the pharmacokinetics and therapeutic outcome of various drugs [22], even if some opposite results were published.

It was reported that in the case of the ABCB1 polymorphism 3435C>T, patients with the CC genotype required a lower CsA dose than carriers of the CT or TT genotypes [17]. This data was corroborated by two meta-analyses conducted among renal transplant recipients, showing that 3435CC carriers required a lower CsA dose to achieve target therapeutic concentrations compared to 3435TT carriers [23,24]. It has also been reported that ABCB1 3435TT carriers had a significantly higher dose-adjusted C0 than the group of CC and CT carriers [25].

In contrast, according to other authors, the basal CsA concentration and dose/concentration ratio were not different among the above-mentioned genotype carriers [14,17]. Moreover, von Ahsen et al. found no effect of the ABCB1 3435C>T polymorphism on dose-adjusted CsA through concentrations or rejection incidence in stable renal transplant recipients [26].

Regarding the other ABCB1 polymorphisms, a recent meta-analysis that focused on the 1236C>T polymorphism [27] showed that there was no significant difference in dose-adjusted C0 between subjects carrying CC and CT genotypes, while a difference was observed for both genotypes compared to TT patients. In a recent meta-analysis of the 1236C>T polymorphism, there was also a difference in dose-adjusted C2 between subjects with CC and CT genotypes.

As regards the ABCB1 2677G>T/A polymorphism, it was shown that patients who were TT carriers had a higher dose-adjusted trough concentration at baseline than the group of GG and GT carriers [26].

On the clinical ground, it is important, to take into account that the 1236TT-2677TT-3435TT haplotype most contributed to the variation in CsA concentrations, as determined by multiple linear regression analysis [11]. This finding is in agreement with earlier observations [26], which suggested ABCB1 haplotype assessment is superior to SNP analysis for predicting CsA blood concentrations [20]. The different effects of the haplotype compared with the single SNPs may also explain the reported discordant results regarding the role of ABCB1 in CsA pharmacodynamics.

#### 3.1.5. NFKB1 Gene

NF-κB is a transcription factor critical for inflammatory responses. It is a key regulator of inflammation binding to the retinoid X receptor (RXR) preventing the SXR-RXR complex from binding to consensus DNA sequences in the regulatory regions of downstream target genes [28].

Despite its role, this gene has received so far poor attention as regards CsA pharmacogenomics. The only available study showed that the genetic make-up at NFKB1-94 influenced CsA pharmacodynamics in patients who were not carriers of the ABCB1 2677 TT and 3435 TT genotypes: subjects with the -94 ATTG ins/ins genotype had a higher dose-adjusted C0 than those with the 94 ATTG del/del genotype [26].

#### 3.1.6. SXR Gene

SXR encodes for an intracellular receptor. When activated by steroid or xenobiotic compounds it forms a heterodimer with RXR and actives the transcription of several genes such as cytochrome P-450 gene cluster and ABCB1 gene.

Transcription of both CYP and ABCB1 genes, and eventually CsA metabolization, is induced by the intracellular receptor encoded by the SXR gene [20]. So, different polymorphisms in the SXR gene can affect transcription and the total activity of metabolizing enzymes. Indeed, it was shown that a deletion in the promoter region of SXR is able to reduce cyclosporine metabolization [29]; furthermore, age was also demonstrated to be able to influence SXR expression [30].

In conclusion, the most concordant data on genetic polymorphisms affecting CsA metabolism regards SXR and ABCB1 polymorphisms. On this basis, the assessment of both polymorphisms before kidney transplantation could allow a more reliable prescription of CsA dose in kidney recipients.

#### 3.1.7. Tacrolimus

Tacrolimus is the other drug member of the calcineurin inhibitor family and is preferred to cyclosporine in high-income countries, due to its fewer side effects and inferior nephrotoxicity, but still, it can increase, as all immunosuppressive therapy, cancer risk, this is one more point to assure the best optimal drug blood concentration.

Despite being part of the same drug family of CsA, it exerts its action in a different way, principally through impairment of gene expression in target cells, namely lymphocytes. Tacrolimus binds to an immunophilin, the FK506 binding protein (FKBP), then the tacrolimus-FKBP complex inhibits the phosphatase activity of calcineurin. The drug is also able to inhibit calcium-dependent events, such as interleukin (IL)-2 gene transcription, nitric oxide synthase activation, cell degranulation, and apoptosis. Finally, tacrolimus is able to inhibit T-cell proliferation in response to ligation of the T-cell receptor [31]. As for CsA, none of the genes involved in the mechanisms of action of tacrolimus seems to be related to its pharmacogenomics.

Similar to CsA, tacrolimus is metabolized mainly by the cytochrome P-450 enzyme, encoded by the CYP3A5 gene, whose polymorphism (CYP3A5*3) was the first described as being able to influence tacrolimus pharmacokinetics.

#### 3.1.8. CYP3A5 Gene

Unlike CsA, all studies of tacrolimus agree on the CYP3A5*3 being the most important polymorphism influencing tacrolimus kinetics, as it encodes a non-active form of the enzyme. As a matter of fact, CYP3A5*3/*3 patients required a significantly lower dosage to achieve therapeutic drug blood concentration [32] than patients carrying the active form of CYP3A5.

A significant clinical result of this discovery was that, by using CYP genotyping as a guide to prescribing drug dosage, it was possible to reduce tacrolimus dosage and avoid tacrolimus overexposure and toxicity in patients with kidney transplantation [33].

#### 3.1.9. CYP3A4 Gene

In patients with CYP3A4 polymorphisms, a wider area under the curve (AUC) of tacrolimus was found [34], due to a lower drug metabolization.

Indeed, tacrolimus clearance has been shown to be significantly associated with the CYP3A4*1B and CYP3A4*22 alleles [34]. These two polymorphisms have the opposite effect. CYP3A4*1B allele was associated with a rapid tacrolimus metabolization in the late post-transplantation period [35] requiring higher dosage compared with wild type allele, while CYP3A4*22 carriers required lower doses of tacrolimus than non-carrier patients [36], this lower metabolization activity is due by the fact that the polymorphism is associated with a lower gene expression.

#### 3.1.10. CYP3A7 Gene 

The only available study of this gene showed that the CYP3A7 rs2257401 polymorphism was positively associated with the adjusted tacrolimus blood concentration/dose ratio, but only in patients expressing the CYP3A5*1 enzyme [37].

#### 3.1.11. Membrane P-glycoprotein—ABCB1 Gene

As regards to the ABCB1 gene, numerous studies have investigated the role of the three main ABCB1 polymorphisms (1236C>T, 2677G>A/T, and 3435C>T) that may affect tacrolimus pharmacokinetics. A few studies have successfully shown an association of tacrolimus pharmacokinetics with the above polymorphisms [38], but others did not reveal any association [37,39]. The absence of homogenous data could be explained by the presence of SXR polymorphisms upstream of the ABCB1 pathway, as described for CsA [29,30].

#### 3.1.12. Other Genes

SNPs influencing tacrolimus C0/dose were also identified, such as IL-3 rs181781 and CTLA4 (Cytotoxic T-Lymphocyte Antigen 4) rs4553808 [40]. Indeed, patients with these polymorphisms required a lower tacrolimus C0/dose ratio, while patients with POR (Cytochrome P450 Oxidoreductase) rs1057868-rs2868177 GC-GT showed a higher tacrolimus blood concentration [41] compared with patients without these recognized polymorphisms.

In conclusion, CYP3A5*3 is the most important genetic polymorphism able to predict tacrolimus kinetics. Therefore, from a clinical point of view, CYP3A5*3 genotyping before kidney transplantation should be carefully considered for the correct prescription of the drug in the first post-operation days.

### 3.2. Inhibitors of Mammalian Target of Rapamycin (mTOR)

#### 3.2.1. Sirolimus

Also known as rapamycin, sirolimus is a macrolide structurally similar to tacrolimus. It is still in use in solid organ transplantation, but without the same wide clinical diffusion [42].

Sirolimus forms a complex with the intracellular protein FKBP12. This complex blocks the activation of the cell-cycle-specific kinase, mTOR. The downstream events that follow the inactivation of mTOR result in the blockage of T cell-cycle progression [43] and T lymphocyte proliferation. In this way, Sirolimus inhibits IL-12, IL-7, and IL-15-driven proliferation of activated T cells. In addition to its effects on T cell activity, it inhibits IL-2-dependent and independent proliferation of human B lymphocytes and soluble CD40L in the mid-G1 phase of the cell cycle [44].

#### 3.2.2. CYP3A5 Gene

The two available studies agree that the CYP3A5 gene polymorphism plays a role in sirolimus pharmacogenomics. Indeed, treatment with sirolimus and low-dose steroids in patients carrying the CYP3A5*1 polymorphism required a significantly higher sirolimus dose to achieve therapeutic blood trough concentrations [45]. Furthermore, patients with the CYP3A5 rs15524 TT polymorphism needed a higher C0/dose than CC patients [46].

#### 3.2.3. Membrane P-glycoprotein—ABCB1 Gene

As regards the ABCB1 gene, CGC/CGC haplotype was more effective than CGC/TTT or TTT/TTT [47] haplotypes at lowering the sirolimus C0/dose ratio.

Regarding the single SNPs, it was observed that the SNP 3435C>T TT reduced C0/dose ratio, but only at the 15th month of treatment [47].

In Caucasian patients, ABCB1 C1236T homozygous mutant TT carriers required a higher sirolimus dose than wild-type GG to achieve target therapeutic concentrations, and the data was confirmed also for the ABCB1 G2677T/A TT genotype.

However, the most recent meta-analysis on this subject [48] was not able to demonstrate a significant association between the ABCB1 C3435T polymorphism and blood C0/dose ratio of sirolimus in renal transplant recipients, but probably because other factors are involved, sirolimus, trough concentrations are only modestly correlated with the area under the curve [49] and if this parameter is the only one considered to study pharmacogenomics it could explain the different results observed.

#### 3.2.4. Everolimus

Everolimus shares the same pharmacogenomics and mechanisms of action as Sirolimus [50] and it is still prescribed in solid organ transplantation.

As regards drug metabolization, there was no association of everolimus pharmacokinetics with CYP2C8 polymorphism after heart transplantation [51], nor with ABCB1, CYP3A5, CYP2C8, and SXR SNPs in renal transplant patients [52].

In conclusion, the literature is too scarce to draw definite conclusions about mTOR drugs class metabolization and the clinical role of mTOR drugs pharmacogenomics. ABCB1 and SXR could be the most promising haplotypes to investigate in future studies of precision medicine.

### 3.3. Glucocorticosteroids

The effects of glucocorticoids are mediated by both genomic and non-genomic mechanisms. Genomic mechanisms implicate the activation or repression of specific genes encoding anti-inflammatory or pro-inflammatory proteins.

The cytoplasmic glucocorticoid receptor/glucocorticoid complex moves into the nucleus, then binds to the DNA or interacts with co-activator complexes, increasing the expression of anti-inflammatory genes and decreasing the expression of pro-inflammatory genes [53]. The complex can also inhibit the activity of the pro-inflammatory transcription factors nuclear factor B (NF-B) and activator protein 1 (AP-1) [54].

As a consequence of the time-consuming mRNA transcription and translation, the genomic glucocorticoid action is characterized by a slow onset (within hours) of the immunosuppressive response [55].

On the other side, non-genomic action is characterized by a rapid onset (seconds to minutes), a short duration of action (60–90 min), and dose dependence [56].

The pharmacogenomics of glucocorticosteroids reflects the complexity of their mechanisms of action.

As reported by Schijvens [53], the results of the pharmacogenomic studies in renal transplantation are generally inconclusive. Anyway, the polymorphisms of glucorticoid receptors seem to play a role: indeed, three polymorphisms in the Glucocorticoid Receptor (GR) gene are known to be associated with reduced sensitivity to both endogenous and exogenous glucocorticoids and may account for a decreased treatment efficacy; these polymorphisms are known as: TthIIII (rs10052957), ER22/23K (rs6189/rs6190), and GR-9β (rs6198). On the other side, two other GR polymorphisms (N363S rs6195 and BC1I rs41423247) are associated with a higher sensitivity to glucocorticoids and an increased drug efficacy [57].

In conclusion, individualized medicine for glucocorticoids is a complex, but very important clinical topic, due to the generalized use of corticosteroids, particularly in the first phase of kidney transplantation. Today, there are several polymorphisms whose assessment could help in prescribing their optimal dosage in kidney transplantation, in order to avoid side effects, as demonstrated in other diseases treated with this class of drugs, such as nephrotic syndrome [58]. Moreover, the importance of the correct dosage is also due to the fact that impairment of growth in young children and delay in puberty commonly present in children receiving glucocorticoids for chronic illnesses such as nephrotic syndrome.

### 3.4. Thiopurines

Thiopurines, particularly azathioprine, were one of the first IS drugs used in kidney transplantation [57]. Thiopurines have a complex metabolism which leads to the formation of 6-thioguanine nucleotides (6-TGN) [59], blocking the consumption of ATP and dATP in DNA synthesis.

Since azathioprine is a well-known drug, characterized by a favorable cost/benefit ratio, it is still used in low- and medium-income countries [52].

#### 3.4.1. NUDT15 Gene

Nudix hydrolase 15 (NUDT 15) is the main gene involved in thiopurine pharmacogenomics and toxicity. The active thiopurine metabolites 6-thio-GTP and 6-thio-dGTP are hydrolyzed by NUDT15. Gene polymorphisms can determine lower enzyme activity, causing higher levels of thiopurine active metabolites: a non-synonymous SNP that causes the change of arginine in position 139 in cysteine was strongly associated with thiopurine-induced early leukopenia [60].

In addition, other polymorphisms in NUDT15 can explain the majority of thiopurine-related myelosuppression in Asians and Hispanics [61].

#### 3.4.2. TPMT Gene

Patients with different Thiopurine S-methyltransferase (TPMT) SNPs can be normal, intermediate, or poor metabolizers of thiopurine, with different levels of drug intolerance. Indeed, TPMT deficiency is the primary genetic cause of thiopurine intolerance [62].

#### 3.4.3. PACSIN 2 Gene

The polymorphism rs2413739 in the protein kinase C and casein kinase substrate in neuron 2 (PACSIN 2) was reported to reduce the activity of azathioprine by reducing DNA synthesis inhibition [62]. However, no clinical application of this data has been reported so far.

In conclusion, new interest has awakened in recent years regarding the pharmacogenomics of this old drug. Considering the concordant data available in the literature and the risk of unpredictable toxicity of azathioprine, it seems reasonable to study NUDT15 and TPMT gene polymorphisms before transplantation.

### 3.5. Mycophenolate Mofetil (MMF)

MMF is the most commonly used antimetabolite drug for kidney transplant immunosuppression in high-income countries. It is an antagonist of inosine monophasphate dehydrogenase (IMPDH) and blocks DNA synthesis by acting on guanine monophosphate (GMP), guanine diphosphate (GDP), and guanine triphosphate (GTP).

#### 3.5.1. UGT Gene

Mycophenolic acid (MPA) pharmacodynamics is regulated by UDP-glucuronosyltransferase (UGT).

In terms of AUC 0-12, the UGT2B7 IVS1 + 985AG polymorphism induced a more active drug metabolism than the AA allele [63]. In terms of dose-adjusted AUC 0-12, UGT 1A7 (622CC) and UGT 1A9-440CT/-331TC polymorphisms were associated with a higher MPA metabolization than all the other polymorphisms, while patients with UGT 1A9-1818T>C and 518C>G polymorphisms were lower metabolizers [64].

#### 3.5.2. MDR2 Gene

The only available study reported a few years ago that the SNP C24T of the multidrug resistance protein 2 (MDR2) reduced MPA AUC, but this effect was lost in the case of co-treatment with CSA [65].

In conclusion, UGT polymorphisms are clearly demonstrated to affect MMF kinetics. Their assessment before kidney transplantation may be useful to prescribe the optimal drug dosage, thus avoiding both high basal MMF levels, which end in toxicity, and low MMF levels, which lead to drug ineffectiveness.

### 3.6. Monoclonal and Polyclonal Antibodies

Cell-based medicinal products (CBMPs) represent a state-of-the-art approach to reducing general immunosuppression in organ transplantation. They were shown to be safe in living-donor kidney transplant recipients, allowing for fewer complications, but failed to show a lower rejection rate when associated with other IS drugs in the first year after renal transplant [66].

Monoclonal antibodies (mAbs) are produced by using identical immune cells of a unique parent cell. They have monovalent affinity, binding the same epitope, while polyclonal antibodies derive from several cell lineages and bind multiple epitopes: chimeric (-ximab) composed of variable regions derived from mice, humans, and other animals; humanized (-zumab) engineered mainly from human and with only a fragment derived from mouse; and fully human (-humab) generated starting from humanized mouse origin.

Differently from the above-mentioned IS agents, mAbs are administered by intramuscular injections, or by intravenous shots or infusion, or subcutaneously, bypassing the digestive system.

Moreover, they are not metabolized by the cytochrome P450 system.

#### 3.6.1. Polyclonal Anti-Thymocyte Globulins (ATG)

Despite polyclonal anti-thymocyte globulin having been used for several years in both organ and hematopoietic cell transplantation as an induction and anti-rejection therapy, individualization of the dose, therapeutic relevance of non-depletive effects, or prediction of long-term effects are still unresolved items [67].

#### 3.6.2. Basiliximab and Daclizumab

Both IS drugs are used in a variety of immune-mediated diseases, such as multiple sclerosis and uveitis, but with limited success [68].

In conclusion, this is a class of drugs not subject to hepatic metabolism. Therefore, individualization of their dosage based on pharmacogenomics is not feasible.

#### 3.6.3. Rituximab

Rituximab, as other drugs able to inhibit lymphocyte action, such as basiliximab and anti-thymocyte globuline (ATG) [69], has been used before transplantation to increase the grade of immunosuppression. The pharmacogenomics of basiliximab and ATG were described above, and the same considerations apply to rituximab.

## 4. Conclusions

As of today, there are more than 250 drugs whose labels include recommendations for genetic analysis [70]. Although it has been demonstrated that determining the dose before the start of immunosuppressive treatment based on specific polymorphism data may reduce overdosage and drug toxicity, patients are rarely genotyped in clinical practice. Rather, continuous adjustments of the IS dose in the first post-op weeks are conducted on the basis of traditional therapeutic drug monitoring.

It is well-known that in the field of transplantation, the first months are critical to avoid rejection episodes and preserve long-term graft function [71]. Therefore, it should be very important to administer the right therapeutic IS dose from the beginning. However, the patient’s body weight is the only tool actually used to prescribe IS doses in clinical practice, even if it is clear that it is not sufficient for the correct prescription of the IS drugs.

Adding genetic screening for the known polymorphisms involved in the specific patient’s IS medication to the knowledge of body weight would permit the identification of (too) slow and (too) fast metabolizers and thus of those cases in which blood drug concentration does not reflect the dose/weight ratio. It is obvious that traditional therapeutic drug monitoring will always be necessary, in the first period after transplantation, as support for the dose assessment of such critical drugs.

Actually, there is still a lot of research to be conducted in the pharmacogenomics of kidney transplantation, even if a not negligible amount of data are available, as summarized in Table 1.

It is clear that CYP3A5*3 can be presently used as the main determinant for tacrolimus dosage, while for other immunosuppressive drugs, such as cyclosporin A, the IS effect is made up of several polymorphisms and SNPs, co-treatment with other drugs, age, and patient gender. About this point, it is important to remark that CYP3A5 polymorphism is involved in hypertension caused by CNI therapy [72]. However, the assessment of SXR and ABCB1 polymorphisms before kidney transplantation could allow a more reliable prescription of CsA dose in kidney recipients. The most recent research has led to promising results for the influence of UGT polymorphisms on MMF blood levels. A more widespread application of this polymorphism at the bedside may allow for the accumulation of clinical pharmacogenetic data to be used for future research.

Actually, genetic screening, especially for several polymorphisms, is an expensive tool; but technology is improving, and the costs of instruments and reactive agents are decreasing; moreover also if involved genes and polymorphisms are numerous, considering a single drug the number of polymorphisms to be identified is more reasonable. Furthermore, not all polymorphisms have a Mendelian distribution, as illustrated in Table 2, so for some populations not all genes and polymorphisms need genetic screening.

## Figures and Tables

**Figure 1 jcm-12-04454-f001:**
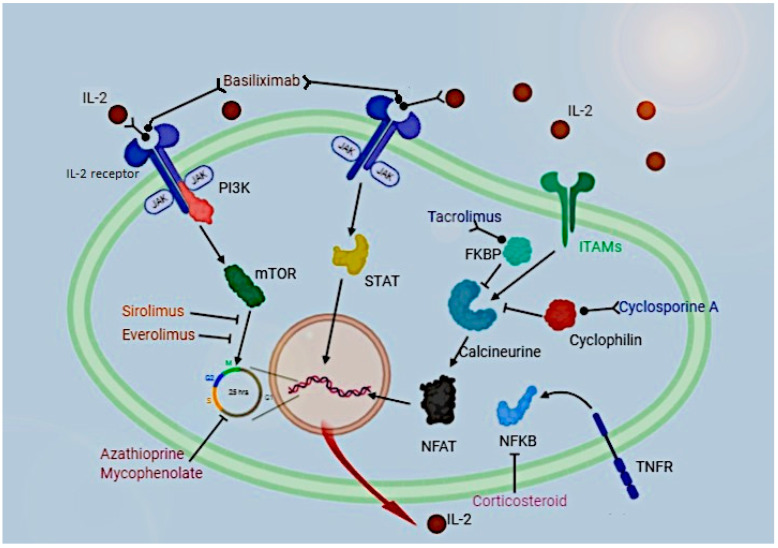
Pathways of action of immunosuppressive drugs in a T-cell.

**Table 1 jcm-12-04454-t001:** Summary of clinically relevant effects reported for the main gene polymorphisms associated with IS drugs for kidney transplantation.

IS Drug	Gene	Polymorphism/Allele	Associated Effect	Reference
Cyclosporine	CYP3A5	* 1	Higher dose in * 1 carrier	[13,14]
	CYP3A4	* 1B	Higher dose in * 1B carrier	[14]
		* 22	Higher C0/dose in * 22 carrier	[20]
	ABCB1	C1236T	Higher dose-adjusted in TT carrier	[26]
		G2677T/A	Higher dose-adjusted in TT carrier	[25,27]
		C3435T	Lower dose in CC carrier	[16,22,23]
			Higher dose-adjusted in TT carrier	[24]
	NFKB1	−94ATTG ins/ins	Higher dose in ins/ins carrier	[25]
	SXR	rs3842689 del/del	Reduces CsA metabolism in del/del carrier	[29]
Tacrolimus	CYP3A5	* 3	Lower dose in * 3 carrier	[31,32]
	CYP3A4	* 1B	Slow drug metabolization in * 1B carrier	[33]
		* 22	Rapid drug metabolization in * 22 carrier	[34,35]
	CYP3A7	rs2257401	Higher dose adjusted in G carrier	[36]
	ABCB1	C1236T; G2677T/A; C3435T;	No consensus	[36,37,38,39,40,41]
	IL-3	rs181781	Lower C0/dose in AA carrier	[39]
	CTLA4	rs4553808	Lower C0/dose in GG carrier	[39]
	POR	rs1057868	Higher blood concentration in T carrier	[40]
		rs2868177	Higher blood concentration in G carrier	[40]
Sirolimus	CYP3A5	* 1	Higher dose in * 1 carrier	[43]
		rs15524	Higher C0/dose in TT carrier	[45]
	ABCB1	C1236T	Higher dose in TT carrier	[47]
		G2677T/A	Higher dose in TT carrier	[47]
		C3435T	Reduced dose in TT carrier	[47]
Glucocorticoids	GR	TTHIIII	Reduced sensitivity in T carrier	[52]
		ER22/23K	Reduced sensitivity in A-A carrier	[52]
		GR-9β	Reduced sensitivity in G carrier	[52]
		N363S	Higher sensitivity in G carrier	[56]
		BC1I	Higher sensitivity in GG carrier	[56]
Thiopurines	NUDT15	p.Arg139Cys	Higher basal level in T carrier	[59]
	TPMT	various	Various effects	[60]
	PCSIN2	rs2413739	Reduced drug activity in C carrier	[61]
Mycophenolate mofetil	UGT2B7	IVS1 + 985AG	Higher drug metabolism in AG carrier	[62]
	UGT1A7	622CC	Higher drug metabolism in CC carrier	[62]
	UGT1A9	−440CT/33TC	Higher drug metabolism in CT-TC carrier	[62]
		−1818T>C	Lower drug metabolism in C carrier	[63]
		518C>G	Lower drug metabolism in G carrier	[63]
	MDR2	C24T	Higher metabolism in T carrier	[64]

**Table 2 jcm-12-04454-t002:** Genetic distribution of considered polymorphism among European, African and Asiatic populations.

GENE	Polymorphism	European	African	Asian
CYP	3A5*3	RA = 0.07 AA = 0.92	RA = 0.69 AA = 0.30	RA = 0.28 AA = 0.71
	3A4 1B	RA = 0.03 AA = 0.96	RA = 0.63 AA = 0.36	RA = 0.00 AA = 1.00
	3A4*22	RA = 0.95 AA = 0.04	RA = 0.99 AA = 0.009	RA = 1.00 AA = 0.00
ABCB1	C1236T	RA = 0.42 AA = 0.57	RA = 0.20 AA = 0.79	RA = 0.62 AA = 0.37
	G2677T/A	RA = 0.44 AA = 0.55/0.001	RA = 0.11 AA = 0.87/0.001	RA = 0.56 AA = 0.43/0.001
	C3435T	RA = 0.51 AA = 0.48	RA = 0.22 AA = 0.77	RA = 0.38 AA = 0.61
NFKB1	−94ATTG	RA = 0.60 AA = 0.39	RA = 0.50 AA = 0.49	RA = 0.64 AA = 0.35
SXR	RS3842689	RA = 0.60 AA = 0.39	RA = 0.71 AA = 0.28	RA = 0.68 AA = 0.31
IL-3	RS1811781	RA = 0.89 AA = 0.10	RA = 0.97 AA = 0.02	RA = 0.68 AA = 0.32
CTLA4	RS4553808	RA = 0.87 AA = 0.12	RA = 0.92 AA = 0.07	RA = 0.96 AA = 0.03
POR	RS1057868	RA = 0.71 AA = 0.28	RA = 0.80 AA = 0.19	RA = 0.61 AA = 0.38
	RS2868177	RA = 0.67 AA = 0.32	RA = 0.62 AA = 0.37	RA = 0.53 AA = 0.46
GR	TTHIIII	RA = 0.68 AA = 0.31	RA = 0.72 AA = 0.27	RA = 0.91 AA = 0.08
	RC22/23K	RA = 0.97 AA = 0.03	RA = 0.98 AA = 0.02	RA = 0.99 AA = 0.01
	GR-9B	RA = 0.82 AA = 0.17	RA = 0.95 AA = 0.05	RA = 0.99 AA = 0.001
	N363S	RA = 0.96 AA = 0.04	RA = 0.99 AA = 0.01	RA = 1.00 AA = 0.00
	BCII	RA = 0.63 AA = 0.36	RA = 0.77 AA = 0.22	RA = 0.79 AA = 0.20
NUDT15	Arg139Cys	RA = 0.99 AA = 0.001	RA = 0.99 AA = 0.001	RA = 0.88 AA = 0.11
PCSIN2	RS2413739	RA = 0.56 AA = 0.43	RA = 0.52 AA = 0.47	RA = 0.87 AA = 0.10
UGT	IVS21 + 985	RA = 0.86 AA = 0.13	RA = 0.72 AA = 0.24	RA = 0.93 AA = 0.07
	622CC	RA = 0.63 AA = 0.36	RA = 0.83 AA = 0.16	RA = 0.90 AA = 0.10
	−440CT	RA = 0.69 AA = 0.30	RA = 0.93 AA = 0.07	RA = 0.97 AA = 0.03
	−1818T>C	RA = 0.75 AA = 0.24	RA = 0.80 AA = 0.20	RA = 0.50 AA = 0.50
	518C>G	RA = 0.77 AA = 0.22	RA = 0.96 AA = 0.04	RA = 0.72 AA = 0.28

RA = reference allele, AA = alternative allele. Data from https://www.ncbi.nlm.nih.gov/snp (accessed on 5 May 2023) for each considered polymorphism. The site consulted on 6 December 2023.

## Data Availability

Data available from authors under reasonable request.

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
