# Peer review of "Pharmacogenomics of Old and New Immunosuppressive Drugs for Precision Medicine in Kidney Transplantation"

_jcm, 2023, doi:10.3390/jcm12134454_

Round 1
Reviewer 1 Report
This is very important issue that gene is very important to metabolize IS drugs. To control therapeutic range of IS drug
I have some suggestions.
1: Gene test is very delicate problem and expensive. What we need for using easily in future?
2: It is easily understanding what types of gene is major in each countries. If you have the data, better to put in this article.
Author Response
Reviewer 1
Comments and Suggestions for Authors
This is very important issue that gene is very important to metabolize IS drugs. To control therapeutic range of IS drug.
I have some suggestions.
1: Gene test is very delicate problem and expensive. What we need for using easily in future?
> Thanks for appreciating the importance of our review. It is true that widespread genetic screening may be expensive, but screening all polymorphisms is not necessary in any individual patient and limiting gene polymorphisms tests to the immunesuppressive agents on use decreases the costs. Moreover, technological advances will succeed in a concomitant cost decrease. In this perspective, we added to the text the following sentence at line 421:
“Actually, genetic screening, especially if extended to many polymorphisms, is an expensive tool, but technology is advancing and the costs of instruments and reactive agents are decreasing. Moreover, even if genes and polymorphisms involved in the metabolism of IS drugs are numerous, considering every single immunosuppressive drug, the number of polymorphisms to be assessed in an individual patient becomes reasonable. Furthermore, since not all polymorphisms have a mendelian distribution, as illustrated in table 2, only a few genes and polymorphisms need genetic screening for the patient of a given population”.
2: It is easily understanding what types of gene is major in each countries. If you have the data, better to put in this article.
> Thanks for this important suggestion. We added a new table with the frequency of the main gene polymorphisms in European, African and Asian populations.
See table 2.
Reviewer 2 Report
This review manuscript focuses on summarizing the use of immunosuppressive drugs in kidney transplantation patients. Overall, this manuscript is too descriptive. The authors basically need to analyze the medication used in the clinic according to published papers and refine the data rather than simply show a drug list. Some introductions for specific genes, such as NFKB1 and SXR, were too simple. The structure of this review manuscript should be reorganized. In addition, the term “pharmacogenomics” in the title might not be appropriate if based on the content of the current manuscript.
The English language requires editing.
Author Response
Comments and Suggestions for Authors
1.This review manuscript focuses on summarizing the use of immunosuppressive drugs in kidney transplantation patients. Overall, this manuscript is too descriptive. The authors basically need to analyze the medications used in the clinic according to published papers and refine the data rather than simply show a drug list.
> Thanks for your criticism. We are sorry if our manuscript is too descriptive, but it was the only way to analyze how every immunosuppressant drug is influenced by genetic polymorphisms. It was not in our intention to analyze the medication used in clinic, but how genetic polymorphisms are able to modify the effect of every medication used in immunosuppressive therapy.
Thanks for your criticism. It is true that some IS medications are currently more utilized than others in clinical practice. However, even the description of those currently less in use may be useful to the reader, as they represent the premise for understanding the development (metabolism?) of the new drugs. We understand that in this way we may increase the impression of a list of drugs, but limiting the number of IS medications on consideration may also carry the risk of incompleteness.
2. Some introductions for specific genes, such as NFKB1 and SXR, were too simple.
>Thanks for you note. We expanded the introduction of SXR and NFKB genes and we added a reference for NFKB regarding how it influences drug metabolism.
For NFKB, the following sentence was added:
NF-κB is a critical transcription factor for inflammatory response. It is a key regulator of inflammation, which binds to the retinoid X receptor (RXR) and prevents the SXR-RXR complex from binding to consensus DNA sequences in the regulatory regions of downstream target genes [28].
For SXR, we added the following one:
SXR encodes for an intracellular receptor. When activated by steroid or xenobiotic compounds, it forms an heterodymer with RXR and activates the transcription of several genes, like cytochrome P-450 gene cluster and ABCB1 gene.
3. The structure of this review manuscript should be reorganized. In addition, the term “pharmacogenomics” in the title might not be appropriate if based on the content of the current manuscript.
>Thanks for your criticism. As regards the structure of the manuscript, we explained in the response to point 1 the reasons which stay behind the decision of describing all the IS drugs in which polymorphisms are able to modify drug metabolism. As regards the use of the term “pharmacogenomics”in the title, the current concept of “pharmacogenomics” is explained in the Introduction section and the review follows this concept.
Reviewer 3 Report
1. Authors write that Cyclosporine is still the most used CNI preparation in low income country. Kindly provide the reference. In South ASIA and Africa, most countries are LMIC, and TAC is the most commonly used CNI.
2. Most commonly drug affected by pharmacogenomics is tacrolimus, however there is little description about CYP3A4 metabolism, there is too much description about cyclosporine.
practically difficult to recommend so many tests, much research is needed before prescribing pharmacogenomics before transplant and it is quite costly.
Poorly written article. Need complete rewriting by English expert writer
Author Response
Reviewer 3
Comments and Suggestions for Authors
1. Authors write that Cyclosporine is still the most used CNI preparation in low income country. Kindly provide the reference. In South ASIA and Africa, most countries are LMIC, and TAC is the most commonly used CNI.
>Thanks for your note, our reference was not so recent. Therefore, we modified the sentence as follows:
While it is no longer used for transplantation in high-income countries, it may still be utilized in low and medium-income countries.
2. Most commonly drug affected by pharmacogenomics is tacrolimus, however there is little description about CYP3A4 metabolism, there is too much description about cyclosporine.
> We agree on your criticism. We described CYP3A4-CsA more than CYP3A4-FK mainly because literature is more discordant for cyclosporine than for tacrolimus. According to your suggestion, we gave more room to tacrolimus and modified the sentence as follows:
In patients with CYP3A4 polymorphisms treated with tacrolimus, a wider area under the curve (AUC) was found [34], due to a lower drug metabolization. Indeed, tacrolimus clearance has been shown to be significantly associated with the CYP3A4*1B and CYP3A4*22 alleles [34]. These two polymorphisms have opposite effects: CYP3A4*1B allele was associated with a rapid tacrolimus metabolization in the late post-transplantation period [35], requiring higher drug dosage compared with the wild type allele, while CYP3A4*22 carriers required lower tacrolimus doses than non-carriers [36]. In these patients, the lower metabolization activity was associated with a lower gene expression.
3.Practically difficult to recommend so many tests, much research is needed before prescribing pharmacogenomics before transplant and it is quite costly.
> Thanks for your criticism. Genetic tests before transplant are expensive, but it is also true that screening for all polymorphisms is not necessary and depends on the drug we want to prescribe. Moreover, technology is increasing with a concomitant cost decrease. In this perspective, we added to the text the following sentence at line 421 and added table 2, which summarizes the polymorphism distribution among different populations:
“Actually, genetic screening, especially if extended to many polymorphisms, is an expensive tool, but technology is advancing and the costs of instruments and reactive agents are decreasing. Moreover, even if genes and polymorphisms involved in the metabolism of IS drugs are numerous, considering every single immunosuppressive drug, the number of polymorphisms to be assessed in an individual patient becomes reasonable. Furthermore, since not all polymorphisms have a mendelian distribution, as illustrated in table 2, only a few genes and polymorphisms need genetic screening for the patient of a given population”.
Reviewer 4 Report
Dear Authors,
Immunosuppressive drugs treatment are important part of kidney transplantation, which suggest need for personalization. Pharmacogenomics factors play crucial role in pharmacokinetics of the main immunosuppressive medications currently used in patients. You are decided to investigate in field of clinical relevant data regarding pharmacogenomic differences, but without PRISMA or any other relevant approach. Maybe you could use PICO approach, define inclusion and exclusion criteria, which will increase the value of your work. I have the impression that you missed important results in this area and did not explain why. You have to give some kind of methods in your review. For example, considering CYP 3A5 polymorphism and tacrolimus, there are novel biomarkers linked to polymorphism. Also, clinical consideration are often in relation with concentration - dose ratio. In this form your results are low.
Author Response
Reviewer 4
You are decided to investigate in field of clinical relevant data regarding pharmacogenomic differences, but without PRISMA or any other relevant approach. Maybe you could use PICO approach, define inclusion and exclusion criteria, which will increase the value of your work.
I have the impression that you missed important results in this area and did not explain why.
You have to give some kind of methods in your review. For example, considering CYP 3A5 polymorphism and tacrolimus, there are novel biomarkers linked to polymorphism. Also, clinical consideration are often in relation with concentration - dose ratio. In this form your results are low.
> Thanks for your note and criticism. As a matter of fact, we applied the PICO approach: our Population and Problem are transplanted patients and immunosoppressive drug metabolization, the Intervention is the treatment with different kinds of immunosuppressive therapy, Comparison is the role of the different polymorphisms and the Outcome is how the patients metabolize each drug. You are right in remarking that this point was not well explicitated, so we have modified the Methods, by adding how we selected and evaluated articles.
Regarding the clinical considerations about drug concentration-dose ratio, the effects of under and overexposure were mentioned in the Introduction, even if in a general way. Now we added to the text more clinical considerations for the main immunosuppressive drugs, such as, at line 423, those regarding the role of CYP3A in inducing hypertension due by calcineurin inhibitor therapy..
Round 2
Reviewer 1 Report
Thanks for adding lines and table due to my suggestion.
Reviewer 2 Report
I have no additional comments.
No.